# Whole blood transcriptional responses of very preterm infants during late-onset sepsis

Sherrianne Ng[1,2,3], Tobias Strunk[4,5], Amy H. Lee[6,7], Erin E. Gill[6,7], Reza Falsafi[6,7], Tabitha Woodman[1,5], Julie Hibbert[5], Robert E. W. Hancock[6,7], Andrew Currie[1,5]*

1 Medical, Molecular and Forensic Sciences, Murdoch University, Perth, WA, Australia, 2 Division of the Institute of Reproductive and Developmental Biology, Imperial College Parturition Research Group, Imperial College London, London, United Kingdom, 3 March of Dimes European Prematurity Research Centre, Imperial College London, London, United Kingdom, 4 Department of Health, Neonatal Directorate, King Edward Memorial Hospital, Child and Adolescent Health Service, Perth, WA, Australia, 5 Neonatal Infection & Immunity Team, Wesfarmers Centre of Vaccine & Infectious Diseases, Telethon Kids Institute, Perth, WA, Australia, 6 Department of Microbiology and Immunology, University of British Columbia, Vancouver, British Columbia, Canada, 7 Centre for Microbial Diseases and Immunity Research, University of British Columbia, Vancouver, British Columbia, Canada

* a.currie@murdoch.edu.au

## Abstract

### Background

Host immune responses during late-onset sepsis (LOS) in very preterm infants are poorly characterised due to a complex and dynamic pathophysiology and challenges in working with small available blood volumes. We present here an unbiased transcriptomic analysis of whole peripheral blood from very preterm infants at the time of LOS.

### Methods

RNA-Seq was performed on peripheral blood samples (6–29 days postnatal age) taken at the time of suspected LOS from very preterm infants <30 weeks gestational age. Infants were classified based on blood culture positivity and elevated C-reactive protein concentrations as having confirmed LOS (n = 5), possible LOS (n = 4) or no LOS (n = 9). Bioinformatics and statistical analyses performed included pathway over-representation and protein-protein interaction network analyses. Plasma cytokine immunoassays were performed to validate differentially expressed cytokine pathways.

### Results

The blood leukocyte transcriptional responses of infants with confirmed LOS differed significantly from infants without LOS (1,317 differentially expressed genes). However, infants with possible LOS could not be distinguished from infants with no LOS or confirmed LOS. Transcriptional alterations associated with LOS included genes involved in pathogen recognition (mainly TLR pathways), cytokine signalling (both pro-inflammatory and inhibitory responses), immune and haematological regulation (including cell death pathways), and metabolism (altered cholesterol biosynthesis). At the transcriptional-level cytokine responses during LOS were characterised by over-representation of IFN-α/β, IFN-γ, IL-1

**Data Availability Statement:** The transcriptomics data presented in this article have been submitted to the NCBI Gene Expression Omnibus under accession number GSE138712.

**Funding:** SN is supported by a NH&MRC of Australia Centre of Research Excellence Scholarship. The work in REWH's lab is supported by the Canadian Institutes for Health Research grant # FDN-154287 and he holds a Canada Research Chair in Health and Genomics and a UBC Killam Professorship. The funders had no role in study design, data collection and analysis, decision to publish, or preparation of the manuscript.

**Competing interests:** The authors have declared that no competing interests exist.

**Abbreviations:** LOS, late-onset sepsis; RNA-Seq, transcriptome sequencing; GA, gestational age; CRP, C-reactive protein; RIN, RNA integrity number; IFN, interferon; IL, interleukin; PPI, protein-protein interaction; MCT4, monocarboxylate transporter 4; BH3, Bcl-2 homology; BAD, Bcl-2-associated death; SREBP, sterol regulatory element-binding proteins; IL1R2, Interleukin-1 receptor type-2; IL1RN, Interleukin-1 receptor antagonist; SOCS, suppressor of cytokine signalling; LPS, lipopolysaccharide; HDL, high-density lipoproteins.

and IL-6 signalling pathways and up-regulation of genes for inflammatory responses. Infants with confirmed LOS had significantly higher levels of IL-1α and IL-6 in their plasma.

## Conclusions

Blood responses in very preterm infants with LOS are characterised by altered host immune responses that appear to reflect unbalanced immuno-metabolic homeostasis.

## Introduction

Neonatal sepsis results in an estimated 0.5–1 million infant deaths globally each year. Preterm infants, particularly those born very preterm (<32 weeks gestation), account for more than a third of the mortality within the first month of life [1–3]. Late-onset sepsis (LOS), with onset after 72h of age, is the most common form of sepsis affecting preterm infants, occurring in more than 20% of those born very preterm. The majority of LOS is due to ubiquitous and often low-virulence commensals of the gastrointestinal tract and skin, especially coagulase negative staphylococci such as *Staphylococcus epidermidis*, suggesting that the preterm immune system may be developmentally immature [4–6].

The preterm infant immune system has both quantitative and functional impairments; with diminished protective function due to anatomical barriers, impaired innate immune defences including reduced pro-inflammatory cytokine production, antigen presentation and phagocytosis of innate cells, and delayed adaptive immune responses due to limited antigenic exposure prior to birth [7–11]. In addition, immune cells of preterm infants have also shown impairments in cellular metabolism that limit innate immune responses during infection [12]. These immune-metabolic deficiencies contribute to an increased risk of preterm infants acquiring infection and ultimately, demonstrating dysregulated inflammatory responses that lead to neonatal sepsis [7, 13–15].

There has been recent interest in understanding neonatal sepsis pathophysiology at the molecular level. Smith *et al* (2014) identified a 52-gene immune-metabolic network that could predict bacterial infections in preterm and term neonates. Cernada and colleagues found 554 differentially expressed genes that discriminated VLBW septic cases from controls (who showed no signs of infection)[16, 17]. Wynn and colleagues found significant differences in transcriptional host responses between patients with early-onset sepsis and LOS, suggesting that accounting for timing of sepsis is critical in transcriptional profiling studies of neonates [18]. However, no study to date has specifically used transcriptome sequencing (RNA-Seq) to investigate immuno-metabolic host transcriptional responses during LOS in very preterm infants, in whom the incidence of sepsis is highest.

In this pilot study, we used RNA-Seq to determine whole blood transcriptional responses of very preterm infants with suspected or confirmed LOS. Our findings are the first to characterise the underlying functional biology of blood leukocyte transcriptional responses associated specifically with LOS immunopathology in very preterm infants.

## Methods

### Ethics statement

The study was approved by the institutional Human Research Ethics Committee at King Edward Memorial Hospital for Women, Perth, Western Australia (HREC reference:

2014091EW). Written informed consent from parents or guardians was obtained prior to study participation.

## Study design

A cohort of 20 very preterm infants (<30 weeks gestational age, GA) was recruited as part of a larger prospective study of 57 infants born <42 weeks GA. Infants were prospectively recruited from the neonatal intensive care unit of King Edward Memorial Hospital, Perth, Australia over a period of 18 months (July 2015 –December 2016). The inclusion criterion for this study was informed consent by parent/guardian for an infant being investigated for suspected sepsis within the first 42 days of life. Relevant demographic (GA, birth weight and postnatal age) and laboratory (blood culture results, serial C-reactive protein (CRP) measurements and differential counts) information of recruited infants was collected from patient records and electronic hospital databases.

Study patients born very preterm (n = 20) were retrospectively classified, based on blood culture results using a recommended minimum volume of 0.5 mL for infants <28 weeks GA and plasma CRP immuno-assay measurements (Vitros, Ortho-Clinical Diagnostics), as having confirmed LOS (positive blood culture and CRP >20mg/L within 72 hours of blood culture), possible LOS (negative blood culture and CRP >20mg/L within 72 hours of blood culture) or no LOS (negative blood culture and CRP <20mg/L within 72 hours of blood culture) [19]. All infants with confirmed LOS received >5 days of antibiotics. Two very preterm infants classified as having a possible blood culture contamination, i.e. those with positive blood culture and CRP <20mg/L, were excluded from further analyses, leaving a final cohort of 18 (Fig 1).

## Blood sampling and processing

Neonatal peripheral blood was collected either at the time of septic screen (n = 13) or within fifteen hours, at the next blood draw for clinical purposes (n = 7, median 10 hours), by venipuncture into a lithium heparin tube, following decontamination of the skin with 70% ethanol and 1% chlorhexidine swab. Blood (0.2–0.5 mL) was stabilized in RNase/DNase-free Eppendorf tubes containing pre-aliquoted PAXgene™ reagent (blood:reagent ratio of 1:2.76) according to manufacturer's guidelines (PreAnalytix; Qiagen/Becton Dickson). Sample tubes were gently inverted ten times and stored at -80˚C until batch RNA extraction. A separate aliquot of blood (0.5 mL) was centrifuged at 3,000$g$ for 20 min at 4˚C to extract plasma, with collected plasma samples stored at -80˚C for immunoassays. All blood samples from very preterm infants were transported on ice to the research laboratory and processed within two hours from blood collection, with the exception of blood samples from two infants with no LOS, which were processed within 7 hours.

## RNA extraction, quantification and quality assessment

RNA was extracted from stabilised blood using an amended version of the PAXgene™ Blood RNA System Kit (PreAnalytix; Qiagen/Becton Dickson). Briefly, samples were thawed at room temperature for up to 2 hours. Samples containing a total blood and PAXgene™ reagent volume of less than 1880μL had RNase-free water added to a final volume of 1880μL to standardise extraction volumes. All samples were centrifuged and had supernatant removed as per manufacturer's instructions. RNase-free water (500μL) was then added to the pellet. All subsequent steps were performed according to the manufacturer's protocol. The RNA concentration and RNA integrity number (RIN) were measured using the Agilent 2100 Bioanalyzer (Agilent Technologies). Samples included in bioinformatic and statistical analyses (n = 18) had total RNA concentrations from 14–262 μg/μL and RIN of 8.3–9.6.

# Study Design

**Fig 1. Study design.** Summary of samples included in bioinformatics and statistical analyses (bolded boxes). Infants were classified as having confirmed late-onset sepsis (LOS), possible LOS or no LOS based on blood culture and CRP measurement results within 72 hours of blood culture.

## RNA-Seq

PolyA-enrichment of total RNA was performed using the NEBNext Poly(A) mRNA Magnetic Isolation Kit (New England Biolabs), followed by cDNA library synthesis using the KAPA Stranded RNA-seq Library Preparation Kit as per the manufacturer's instructions (Kapa Biosystems). Quantification of the cDNA libraries was performed using a Quant-iT dsDNA Assay Kit (Invitrogen) and normalized to 4 nM. To derive the transcriptome for each infant, samples were labelled with a unique barcode, multiplexed, and sequenced on a HiSeq 2500 sequencer (Illumina), using the High Output 100-bp single-end run, at the University of British Columbia Sequencing Centre.

The deconvoluted sequences from each infant were analysed and the quality of FASTQ reads were assessed using FastQC(v0.0.15), and summarized using MultiQC(v0.8.dev0) [20, 21]. FASTQ reads were aligned to the GRCh37(hg19) human reference genome [22] using the STAR(v2.5) aligner, followed by generation of read counts using htseq-count(v0.6.1p1) [23].

Globin transcripts and low count genes were bioinformatically removed from the count matrix. The median library size of uniquely mapped reads for very preterm infant samples was 6.8 million reads, and a range of 4.1–8.9 million reads for the entire study.

## Cytokine immunoassays

The concentrations of interferon (IFN)-α, IFN-β, IFN-γ, interleukin (IL)-1α, IL-1β, IL-10 and IL-6 in plasma were measured using ProcartaPlex™ immunoassay kits (Invitrogen, Thermo Fisher Scientific) according to the manufacturer's instructions.

## Bioinformatics and statistical analysis

Bioinformatics and statistical analyses were performed on the 18 samples included in the final study design (Fig 1). Descriptive statistics were conducted in PRISM 7 (GraphPad Software Inc., California, USA) and included the Mann-Whitney test to determine statistical differences between clinical groups. All packages used for bioinformatics analyses were performed in R (v3.3.1) [24]. Differential gene expression analysis was conducted using DESeq2 (v1.12.4), whereby counts were normalised using variance stabilising transformation then visualised using principal component analysis [25] with plots generated using ggplot2 (v2.2.1) [26]. Differentially expressed genes, based on pair-wise comparison between condition groups, were identified based on an adjusted p-value of ≤0.05 and ±1.5 fold-change. Statistically significant pathways with Bonferroni-corrected hypergeometric test p-value <0.05 were identified from each differentially expressed gene list using Sigora (v2.0.1) using Reactome database annotations [27] based on over-represented gene-pair signatures. NetworkAnalyst [28] was used for protein-protein interaction (PPI) based network visualisations of gene expression changes based on the InnateDB interactome database [29]. NetworkAnalyst was also used to identify pathway enrichment based on Reactome database pathway and reaction annotations (www.reactome.org) [27, 28]. Sub-networks were extracted based on significantly enriched pathways and were grouped into functional categories. BiomaRt(v2.28.0) was used to convert Ensembl gene identifications to gene symbols [30, 31].

## Results

### Patient demographics

The demographic information of very preterm infants in this study is shown in Table 1. Overall, infants from all clinical groups were similar with no significant differences in GA, birth weight or postnatal age (p>0.05) (S1 Table). Infants with confirmed LOS were infected with either Gram-positive (*Staphylococcus capitis* and/or *Staphylococcus epidermidis*) or Gram-negative (*Escherichia coli* or *Enterobacter asburiae*) pathogens (Table 1).

### Differential gene expression and cell counts between clinical groups

We conducted RNA-Seq to determine possible differences in gene expression between infants with confirmed LOS, possible LOS and no LOS. To determine the major factors contributing to overall differences principal component analysis was first conducted on normalised gene count data (Fig 2). This demonstrated that 4 of the 5 infants with confirmed LOS had similar overall transcriptional responses that were distinct from responses in infants with possible or no LOS. Of note, the single infant with confirmed LOS who did not cluster with the other 4 confirmed LOS patients was distinct based on the presence of polymicrobial infection and a higher postmenstrual age of 237 days at the time of sample collection (Table 1).

**Table 1. Demographics of very preterm infants in each analysis cohort with confirmed possible and no Late-Onset Sepsis (LOS); with additional details on confirmed LOS infants.**

| Variable | Median (Range) | | |
|---|---|---|---|
| | Confirmed LOS; n = 5 | Possible LOS; n = 4 | No LOS; n = 9 |
| GA (weeks)* | 26.9 (24.3–29.9) | 26.4 (24.4–27.0) | 26.4 (23.7–29.8) |
| Birth weight (grams)* | 1286 (620–1650) | 645 (560–790) | 900 (575–1340) |
| Postnatal age (days)* | 11 (6–37) | 21 (14–29) | 12 (6–25) |
| Sex (male) | 4/5 | 3/4 | 4/9 |

**Septic infants**

| Infant ID | Gestation (weeks) | Birth weight (g) | Sex | Postnatal age (days) | Postmenstrual age (days)† | Infecting organism |
|---|---|---|---|---|---|---|
| 5 | 29.9 | 1650 | M | 11 | 220 | *S. capitis* |
| 12 | 28.6 | 1340 | M | 37 | 237 | *S. epidermidis & S. capitis* |
| 16 | 24.3 | 620 | F | 9 | 179 | *E. coli* |
| 17 | 24.6 | 770 | M | 14 | 186 | *S. aureus* |
| 18 | 26.9 | 1286 | M | 6 | 194 | *E. asburiae* |

*Median (Minimum—Maximum).

†Gestation plus postnatal age.

Differential gene expression analysis was subsequently performed to identify genes that were significantly different between clinical groups using as cut-offs an adjusted P-value of ≤0.05 and ±1.5 fold-change. We found that the transcriptional responses of infants with confirmed LOS were substantially different from those with no LOS, with 1,317 differentially expressed genes (995 up-regulated and 322 down-regulated; Fig 2 and S2 Table). In contrast, the differences in transcriptional responses of infants with confirmed LOS and possible LOS were markedly lower, with only 21 differentially expressed genes identified (19 up-regulated and 2 down-regulated; S3 Table). Interestingly, there were no differentially expressed genes identified when comparing transcriptional responses of very preterm infants with possible LOS or no LOS. Analysis of differential counts for white cells, platelets, red cells, neutrophils, lymphocytes, monocytes and eosinophils (S1 Table) showed that there were no significant differences (p>0.05) in cell proportions between the clinical groups except for platelet count between infants with no LOS and confirmed LOS (p = 0.01; S4 Table).

## Transcriptional changes during confirmed LOS

Comparing infants with confirmed LOS to those without LOS, a biological pathway over-representation analysis was performed using SIGORA on all differentially expressed genes. In contrast to traditional pathway enrichment analysis, where pathways are classified as sets of genes with individual genes being equally informative, gene-pair signature pathway analysis accounts for only statistically significant gene-pairs that, taken together, are unique to a single pathway. This reduces the risk of identifying redundant pathways (in which the same genes appear repeatedly in multiple pathways) and allows more appropriate examination of pathways associated with disease [27]. Overall, 39 pathways that function in a broad range of host responses, were significantly (p<0.05) altered in infants with confirmed LOS (Table 2). Of these over-represented biological pathways, 31 comprised only up-regulated genes, 1 had only down-regulated genes and 6 included both up- and down-regulated genes. To better understand the relationship between these pathways and their role during confirmed LOS in very preterm infants, we investigated these pathways based on their involvement in key processes involved in host responses to infection, namely: pathogen recognition, cytokine signalling, immune/haematological

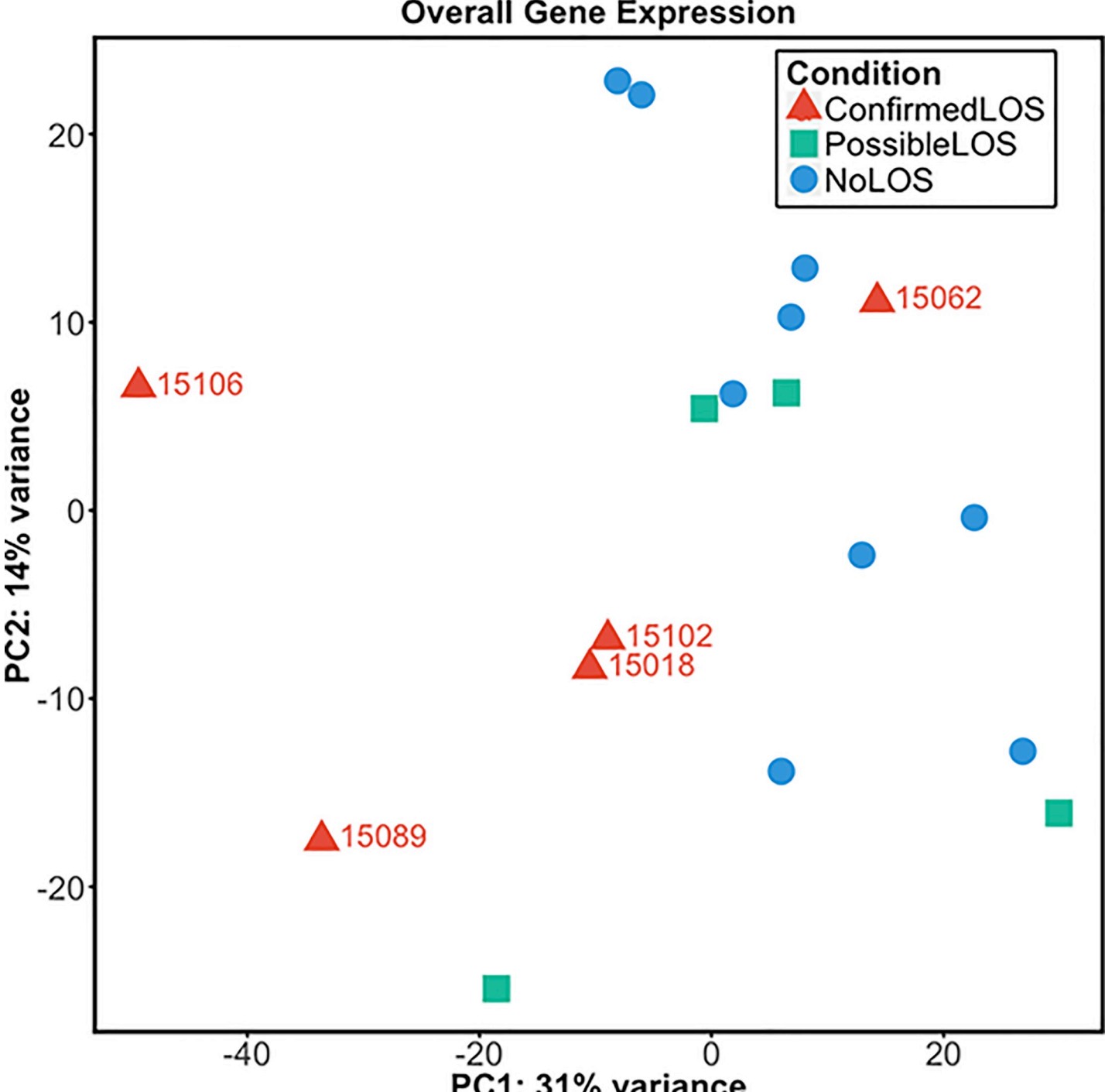

**Fig 2. Principal component analysis of global gene expression.** The graph shows the first two principal components based on overall gene expression between very preterm infants with confirmed late-onset sepsis (LOS; red triangles), possible LOS (green squares) or no LOS (blue circles). The number of DEGs between infants with confirmed LOS/no LOS, 1317; confirmed LOS/possible LOS, 21; and possible LOS/no LOS, 0.

regulation, and metabolism. For each of these key processes, expression was investigated and visualised by plotting the underlying interactions between the protein products of these genes, by generating zero-order PPI sub-networks using NetworkAnalyst (Fig 3).

**Table 2. Functional classification of statistically significant over-represented pathways (determined using the Sigora gene-pair signature method) in infants with Late-Onset Sepsis (LOS).**

| Pathway description | P value[†] | Differentially expressed genes in pathway |
|---|---|---|
| **Pathogen/Pattern Recognition** | | |
| Toll Like Receptor TLR6:TLR2 cascade | 2.82E-20 | Up-regulated |
| Toll Like Receptor TLR1:TLR2 cascade | 5.03E-08 | Up-regulated |
| MyD88-independent TLR3/TLR4 cascade | 5.12E-06 | Up-regulated |
| MyD88:Mal cascade initiated on plasma membrane | 1.24E-05 | Up-regulated |
| Toll Like Receptor 3 (TLR3) cascade | 4.25E-05 | Up-regulated |
| TRIF-mediated TLR3/TLR4 signalling | 1.29E-02 | Up-regulated |
| Toll Like Receptor 9 (TLR9) cascade | 2.86E-02 | Up-regulated |
| Dectin-2 family | 8.20E-04 | Up-regulated |
| **Cytokine Signalling** | | |
| Interferon alpha/beta signalling | 5.33E-27 | Up-regulated |
| Interferon signalling | 8.20E-19 | Up-regulated |
| Interleukin-6 signalling | 2.12E-12 | Up-regulated |
| Interleukin-1 signalling | 2.86E-06 | Up-regulated |
| Negative regulators of RIG-I/MDA5 signalling | 7.40E-06 | Up-regulated |
| Interferon gamma signalling* | 7.48E-05 | Up-regulated |
| Signalling by interleukins* | 7.57E-05 | Up-regulated |
| **Immune/Haematological Regulation** | | |
| Alternative complement activation | 1.32E-03 | Up- and down-regulated |
| **Hemostasis** | | |
| Platelet degranulation | 2.34E-11 | Up-regulated |
| Cell surface interactions at the vascular wall | 3.31E-03 | Up-regulated |
| **Signal transduction** | | |
| RHO GTPases activate WASPs and WAVEs | 2.05E-03 | Up-regulated |
| Signalling to RAS | 5.02E-03 | Up-regulated |
| Hedgehog "on" state | 4.74E-03 | Down-regulated |
| **Cellular responses to external stimuli** | | |
| Diseases of immune system | 1.68E-08 | Up-regulated |
| Oxygen-dependent proline hydroxylation of Hypoxia-inducible Factor Alpha | 1.32E-04 | Up-regulated |
| **Programmed cell death** | | |
| Activation of BAD and translocation to mitochondria | 3.22E-03 | Up-regulated |
| Activation of BH3-only proteins | 1.74E-02 | Up-regulated |
| **Metabolism** | | |
| Activation of gene expression by SREBF (SREBP) | 1.23E-04 | Up-regulated |
| Regulation of cholesterol biosynthesis by SREBP (SREBF) | 2.07E-02 | Up-regulated |
| Cholesterol biosynthesis | 3.23E-02 | Up-regulated |
| Metabolism of vitamins and cofactors | 1.60E-02 | Up- and down-regulated |
| PPARA activates gene expression | 2.95E-02 | Up- and down-regulated |
| **Metabolism of proteins** | | |
| O-linked glycosylation of mucins | 6.02E-15 | Up-regulated |
| **Metabolism of RNA** | | |
| KSRP (KHSRP) binds and destabilizes mRNA | 6.20E-03 | Up-regulated |
| **Developmental Biology** | | |
| Sema4D in semaphorin signalling | 8.65E-04 | Up- and down-regulated |
| Constitutive signalling by NOTCH1 HD domain mutants | 4.96E-03 | Up- and down-regulated |

[†]Statistical significance was based on the hypergeometric test with Bonferroni correction.

## Pathogen recognition and cytokine signalling

Many transcriptional responses in LOS patients were associated with TLR signalling pathways (Table 2). The zero-order PPI sub-network of these functionally enriched TLR pathways showed up-regulation of key interacting genes involved in innate immune signalling including *NFKBIA*, *MYD88*, *CEBPB*, *STAT1*, *IRF7*, *IRAK2*, *IRAK4* and *TBK1* (Fig 3A), which are known to drive the production of pro-inflammatory cytokines and type I IFNs [17, 32]. Genes from the dendritic cell-associated C-type lectin-2 (Dectin-2) family of CLR were also up-regulated (Table 2 and S5 Table), although the CLR signalling pathway was not itself functionally enriched in the zero-order PPI network.

With regards to cytokine responses in LOS patients, it was found that the IFN-α/β, IFN-γ, IL-1 and IL-6 pathways were over-represented, with 128 up-regulated and 13 down-regulated nodes in a large cytokine signalling sub-network (Table 2 and Fig 3A). In addition, immune inhibitory signalling genes for IL-1 receptor antagonist proteins (*IL1R2* and *IL1RN*) and suppressor of cytokine signalling proteins (*SOCS1* and *SOCS3*) were also up-regulated. Interestingly, *SOCS1* was part of the unique gene-set associated with all IFN signalling pathways, whereas *SOCS3* was part of the gene-set for the IL-6 signalling and IFN signalling pathways (S5 Table).

## Immune and haematological processes

Transcriptional regulation of immune and haematological processes during confirmed LOS occurred on multiple levels. Significantly over-represented pathways were identified that were involved in haemostasis, signal transduction, cellular responses to external stimuli and programmed cell death (Table 2 and S5 Table). All pathways comprised up-regulated genes, except for the alternative complement activation (up- and down-regulated genes) and hedgehog "on" state (down-regulated genes) pathways. It was also found that the oxygen-dependent proline HIF1A pathway, commonly associated with hypoxia, was over-represented, with up-regulation of *HIF1A* in the immune/haematological PPI sub-network of differentially expressed genes associated with LOS (Fig 3B). The gene for lactate dehydrogenase A (*LDHA*), which is induced by HIF1α, was also up-regulated along with *SLC16A3*, the gene coding for monocarboxylate transporter 4 (MCT4) [33].

Interestingly, pathways associated with programmed cell death were also identified with the over-representation of the Bcl-2 homology (BH3) protein and Bcl-2-associated death (BAD) pathways (Table 2). Importantly, our immune/haematological PPI sub-network revealed that despite up-regulation of the majority of genes in the network, *BCL2* and *IL-7R*, two key genes involved in controlling apoptosis, were significantly down-regulated (Fig 3B).

## Changes in metabolic pathways

Significant transcriptional changes during confirmed LOS were identified that were associated with lipid metabolism and more specifically, activation and regulation of cholesterol biosynthesis (Table 2). The relationship between cholesterol metabolism, type I IFNs and suppression of IL-1 has previously been elucidated [34, 35], while paediatric sepsis patients also have major alterations in lipid metabolism [36]. We therefore explored gene expression and interaction during metabolism by generating an immuno-metabolism sub-network consisting of functionally enriched genes and pathways involved in the regulation of cholesterol biosynthesis, cytokine signalling and pathogen/pattern recognition (Fig 3A). The sterol regulatory element-binding proteins (SREBP) pathway, involved in responding to low concentrations of cholesterol and subsequent promotion of cholesterol biosynthesis [37, 38], was over-represented along with significant up-regulation of cholesterol biosynthesis genes *SREBF2*, *DHCR7*,

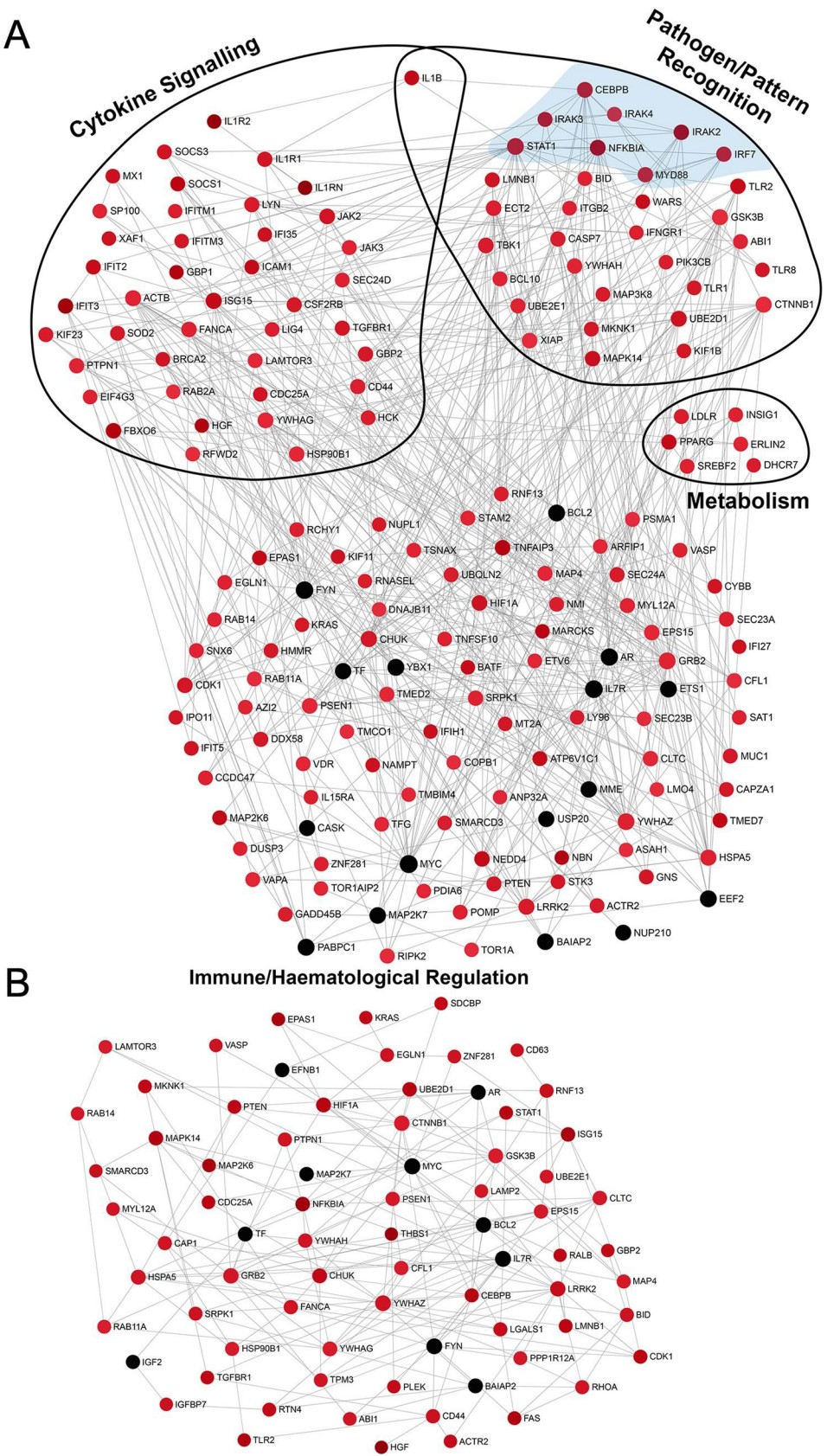

**Fig 3. PPI network visualisations associated with four main functions transcriptionally altered in infants with confirmed Late-Onset Sepsis (LOS).** Zero-order networks were generated in NetworkAnalyst from differentially expressed genes comparing very preterm infants with confirmed LOS to those with no LOS. The sub-networks shown were extracted based on pathways identified from the over-representation analysis (Table 2) that were associated with (A) pathogen/pattern recognition with innate immune signalling nodes highlighted in blue, cytokine signalling and metabolism, and (B) immune/haematological regulation. The nodes were colour coded based on up-regulation (red) or down-regulation (black).

*INSIG1*, *ERLIN2*, *SQLE*, *IDI1* and *LDLR* (Fig 3, Table 2 and S5 Table). Immune-metabolic regulatory genes, *PPARG* (encoding the proliferator-activated receptor gamma protein, a nuclear receptor involved in controlling beta-oxidation of fatty acids and regulation of glucose metabolism) [39, 40] and *CEBPB* (a transcription factor important for regulating immune and inflammatory response genes) were also up-regulated. Importantly, this metabolism sub-network indicated connections between cholesterol biosynthesis genes (i.e. *SREBF2*, *INSIG1* and *ERLIN2*), regulatory genes (i.e. *PPARG* and *CEBPB*) and genes involved in the pro-inflammatory response (e.g. *IL1B*).

## Validation of transcriptional responses in infants with confirmed LOS

To validate the transcriptional findings associated with over-represented cytokine signalling pathways (Table 2), we measured plasma concentrations of IFN-$\alpha$, IFN-$\beta$, IFN-$\gamma$, IL-1$\alpha$, IL-1$\beta$ and IL-6 at the protein-level (Fig 4). The concentrations of all IFNs were below detection limits. However, consistent with our findings at the transcriptional-level, infants with confirmed LOS had significantly higher levels of IL-1$\alpha$ and IL-6 ($p<0.05$) compared to infants with no LOS. Infants with confirmed LOS did not have significantly different levels of IL-1$\beta$ compared to infants with no LOS (median of 5.19 pg/mL in confirmed LOS versus 0.87 pg/mL in no LOS; $p = 0.14$). However, comparison between 4 of the 5 infants with confirmed LOS that had similar overall transcriptional responses (Fig 2) showed significantly higher levels of IL-1$\beta$ compared to all infants with no LOS (median of 9.03 pg/mL in confirmed LOS versus 0.87 pg/mL in no LOS; $p<0.05$). Infants with confirmed LOS also had up-regulated expression of *S100A12*, which encodes for the S100A12 alarmin previously implicated during neonatal and adult sepsis [41–43]. In line with our differential gene expression findings, the IL-1$\alpha$ and IL-1$\beta$ levels of possible LOS infants were similar to those in infants with no LOS. However, infants with possible LOS also had significantly higher IL-6 protein concentrations than infants without LOS, consistent with the elevated CRP levels in these infants [44, 45]. Plasma concentrations of IL-10 were significantly higher in infants with confirmed ($p<0.005$) or possible LOS ($p<0.05$) compared to those with no LOS. There were however no significant differences in IL-10 or IL-6-related gene expression in infants with possible LOS compared to those with no LOS. Although the IL-10 signalling pathway was not significantly over-represented in our pathway analysis, infants with confirmed LOS had up-regulated expression of *IL10* compared to no LOS infants.

## Discussion

Very preterm infants are at highest risk of acquiring LOS, however, our understanding of their underlying leukocyte transcriptional responses during sepsis remains limited. Using a combined approach of differential gene expression, pathway over-representation and network analyses, RNA-Seq was employed to examine the whole blood transcriptional responses of very preterm infants with confirmed or suspected LOS. Our findings showed that: (i) infants with confirmed LOS had significantly altered regulation of host immune responses at the transcriptional-level and significantly higher IL-6 and IL-10 plasma concentrations compared to

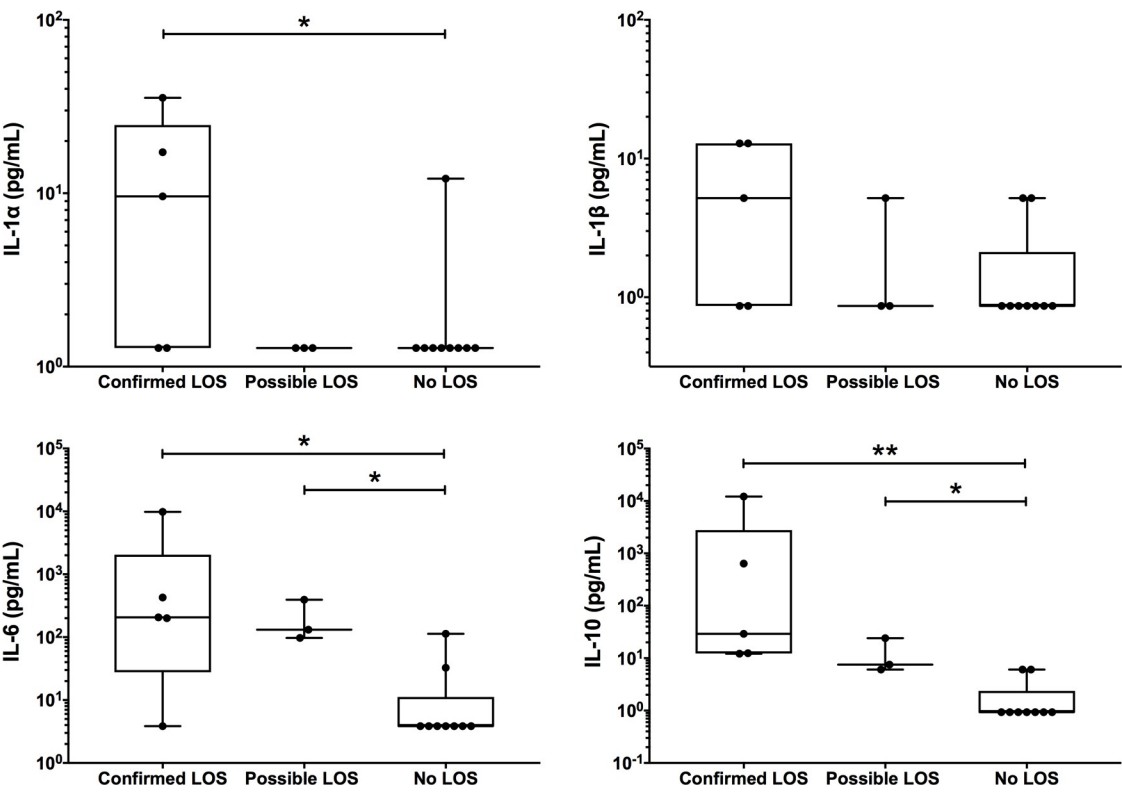

**Fig 4. Cytokine expression of IL-1α, IL-1β, IL-6 and IL-10.** Data represent the concentrations of cytokines IL-1α, IL-1β, IL-6 and IL-10 in infants with confirmed late-onset sepsis (LOS), possible LOS and no LOS. The box plots show the median and interquartile range and statistically significant differences based on the Mann-Whitney Test (*, $p < 0.05$ and **, $p < 0.005$).

infants without LOS; (ii) infants with confirmed LOS had changes in immuno-metabolism that were associated with LOS pathophysiology; and (iii) infants with possible LOS had no differentially expressed genes compared to infants without LOS at the transcriptional-level but had significantly higher IL-6 and IL-10 plasma concentrations at the protein-level.

Transcriptional profiling of blood leukocyte responses revealed that infants with confirmed LOS, when compared to infants without LOS, had 1,317 differentially expressed genes associated with 39 over-represented biological pathways that were involved in a broad range of biological processes from pathogen recognition to metabolism. We found that both immuno-stimulatory and inhibitory transcriptional responses occurred simultaneously during confirmed LOS, indicative of the dynamic pathophysiology of sepsis that is not marked by an initial hyper-inflammatory response followed by a later immunosuppressive anti-inflammatory phase [15, 46, 47]. Immune stimulatory responses included up-regulated genes involved in pro-inflammatory responses including *MYD88*, *STAT1*, *IRF7*, *NFKBIA* and *IL1R1*, which are associated with over-represented cytokine signalling pathways for IFN-α/β, IL-1 and IL-6. These genes were also found to be increased in expression during neonatal infection in a transcriptional study of preterm and term infants [17]. However, although the IFN-α/β, IL-1 and IL-6 signalling pathways were over-represented and consisted of only up-regulated genes, cytokine immunoassays showed that at the protein-level, only IL-1α and IL-6 concentrations were significantly higher in confirmed LOS compared to no LOS infants. This is consistent with the concept that the type I interferon signalling pathways invoke other host defence mechanisms other than just interferons [48].

The higher IL-6 plasma concentrations in our study are in line with previous studies of preterm infants that showed increased plasma IL-6 concentrations during sepsis compared to non-infected or control groups [45, 49, 50]. Separately, IL-1 signalling is mediated by both IL-1α and IL-1β binding to the IL-1R1 receptor. Although IL-1β has been implicated during neonatal sepsis, less is known about IL-1α in human neonates [51–54]. The elevated IL-1α levels in our study were consistent with a recent murine study that also showed increased IL-1α during polymicrobial sepsis [51]. The study identified that IL-1α, and not IL-1β, was responsible for IL-1R1-dependent lethality during neonatal sepsis. In keeping with this, although we found up-regulation of *IL1B* in infants with confirmed LOS, this did not translate to significant differences in IL-1β plasma concentrations compared to infants with no LOS. Our findings were consistent with other neonatal studies that found IL-1β concentrations were not different during sepsis [55–57]. However, our findings differed from studies that reported significantly elevated IL-1β plasma and serum levels in infected compared to non-infected infants [53, 54, 58], except when comparing the 4 of 5 infants with confirmed LOS that had similar transcriptional responses to all infants with no LOS. Overall, the discrepancies between our study and previous studies [53, 54, 58] could be due to multiple factors including different laboratory methods used, sample sizes, sample collection times and neonatal populations.

Importantly, we found up-regulation of *IL1R2* and *IL1RN* in infants with confirmed LOS, with these genes directly connected to *IL1B* in the cytokine signalling sub-network. These results were consistent with two independent transcriptional studies showing up-regulation of *IL1R2* and *IL1RN* during sepsis in preterm/term infants [17] as well as very preterm infants [16] with a role in immune inhibitory signalling. These genes encode for the IL-1 receptor type-2 (IL1R2) and IL-1 receptor antagonist proteins (IL1RN), where IL1R2 acts as a decoy receptor that competitively binds to IL-1α/β and preventing binding to IL1R1, whereas IL1RN prevents IL-1 from binding to the IL1RAP receptor, inhibiting IL-1 signalling [17, 59]. These findings suggest that IL-1β plasma concentrations may not have increased during confirmed LOS in our study due to negative regulation by both IL1R2 and IL1RN.

In parallel with the induction of pro-inflammatory cytokine responses, we also found up-regulation of anti-inflammatory *IL10*, whereby plasma IL-10 concentrations were significantly higher in infants with confirmed LOS compared to infants with no LOS. This was consistent with previous studies where preterm infants with sepsis had significantly increased IL-10 levels compared to those who were non-infected [45, 60]. In addition to *IL10*, we showed up-regulation of *SOCS1* and *SOCS3*, which form part of the IFN signalling pathways. This finding concurs with a previous study showing that septic preterm/term infants had up-regulated *SOCS1* that functioned in immune inhibitory signalling [17]. *SOCS1* and *SOCS3* code for the suppressor of cytokine signalling (SOCS) family of proteins that play a critical role in the maintenance of immune homeostasis [61]. SOCS proteins, which can be induced by cytokines (e.g. IL-10) and bacterial products (e.g. lipopolysaccharide, LPS), negatively regulate the signalling of cytokines, such as type I IFNs, through direct inhibitory modulation of cytokine receptors via the JAK-STAT pathway. This in turn has profound regulatory effects on inflammatory responses and immunity that affects a range of immune cells including T cells, macrophages and neutrophils [48, 62–65]. Although the direct mechanisms of the SOCS family of proteins have not been established for neonatal sepsis, our findings suggest a potential role for SOCS1 and SOCS3 in dampening interferon responses. In particular, IL-10 may inhibit IFN-γ signalling and subsequent IFN-γ production via SOCS1, resulting in undetectable IFN-γ protein. The role of SOCS1 and SOCS3 for immunomodulation during neonatal sepsis warrants further investigation.

Another marker of sepsis in both neonates and adults that was up-regulated here is S100A12 (also called calgranulin C), which is an alarmin involved in regulating innate

immune/inflammatory responses [41, 42]. S100A12 is overexpressed during inflammation and might serve as a general marker of inflammatory diseases, as well as one of the markers of sepsis immunosuppression through endotoxin tolerance [43].

Overall, our findings are consistent with an emerging view describing neonatal sepsis pathophysiology as being characterised by simultaneous hyper- and hypo-inflammatory immune responses [15] and suggest that impaired immuno-regulation may be a key feature of LOS in very preterm infants.

The regulation of immune responses and maintenance of immune homeostasis also depends on tight control of programmed cell death, an essential cellular self-destruction mechanism to ensure tissue homeostasis and elimination of damaged and/or infected cells [66]. In confirmed LOS, we found down-regulation of *BCL2* and *IL-7R*, two key genes important for controlling apoptosis [67, 68]. In two separate mouse models, the over-expression of BCL2 during sepsis was protective against the death of both lymphoid (thymocytes and splenic T cells) or myeloid cells (dendritic cells and macrophages), and was important for host immune responses during sepsis [69, 70]. Accordingly, the significant down-regulation of BCL2 could potentially contribute to increased cell death of immune cells important for host defences during confirmed LOS. Our study did not include examination of the frequencies of the cell populations in blood, however, based on our transcriptional findings and the presently limited understanding of BCL2 for sepsis pathophysiology [71], investigation of the effects of BCL2 for regulation of immune cell death during neonatal sepsis is warranted. We also found down-regulation of *IL-7R*, the receptor important for mediating the activity of IL-7, a hematopoietic growth factor crucial for immune system development and lymphocyte survival [68, 72]. In experimental models of sepsis, IL-7 was responsible for regulating BCL2 to block lymphocyte apoptosis, restoring IFN-γ production and improving recruitment of immune cells to the infection site [73, 74]. Further, lymphocyte depletion has been reported in post-mortem EOS and LOS studies of preterm and term infants, with results suggesting that severe neonatal sepsis could be associated with sepsis-induced immune cell apoptosis [15, 75–77]. Therefore, the disruption of IL-7/IL-7R signalling during confirmed LOS has potentially detrimental implications for regulation of immune and cellular homeostasis that contributes to sepsis pathophysiology. The potential of using IL-7 as a lymphostimulating therapy for septic patients has been described in previous studies [73, 74] and could also be explored in the context of neonatal sepsis. We found an intersection between host immune responses and metabolism at the transcriptional level in infants with LOS. Infants with confirmed LOS had multiple changes in gene expression and pathways associated with cholesterol metabolism and biosynthesis. Several genes identified as up-regulated (*SQLE, IDI1, DHCR7* and *LDLR*) in infants with confirmed LOS were consistent with a study of preterm and term infants that also found transcriptional changes to these genes involved in cholesterol biosynthesis and homeostasis during neonatal sepsis [17]. Specifically, *DHCR7, IDI1* and *LDLR* were also found to be up-regulated during bacterial sepsis in VLBW infants [16]. The cross-regulation between metabolism and the immune system is increasingly recognised as important for host responses during infection [13, 17, 78].

Numerous studies in both humans and mouse models have explored the role of cholesterol, both high-density lipoproteins (HDL) and low-density lipoproteins, in relation to sepsis [79–86]. Adult patients with severe sepsis typically have lower cholesterol levels, and high cholesterol levels are suggested to be important for protection during sepsis, whereby lipoproteins play a crucial role in regulating immune responses to infection such as controlling pro-inflammatory cytokine release, exerting anti-inflammatory effects by neutralising LPS and facilitating clearance of bacterial toxin [79, 84, 87]. Our study did not measure total cholesterol, HDL or low-density lipoproteins levels, however, LOS has been linked to lower total cholesterol and

HDL levels in preterm and term infants, with impaired LPS neutralisation and effects of cytokines suggested to contribute to reduced HDL levels [86, 88].

The metabolism sub-network identified demonstrated connections between metabolic (*SREBF2*, *INSIG1* and *ERLIN2*), regulatory (*PPARG* and *CEBPB*) and inflammatory (*IL1B*) genes, which were all up-regulated, suggesting potential inflammasome activation by sterols during confirmed LOS in our study [38]. Acute infections and sepsis have been linked to diminished ability of HDL to mediate cholesterol efflux from macrophages to plasma [38, 89], which could explain lower total plasma cholesterol and HDL levels during LOS reported in a preterm and term infant study [86]. Specifically, this leads to cholesterol accumulation in macrophages, which promotes inflammasome activation that leads to IL-1β production, and that drive inflammatory responses during microbial infection [34, 35, 38]. Although altered cholesterol homeostasis could contribute to pro-inflammatory responses, we did not find significantly increased plasma IL-1β concentrations in infants with confirmed LOS compared to infants with no LOS at the protein-level. In accordance with a study of neonatal monocytes, this could be due to low expression of *NLRP3* in very preterm infants, where NLRP3 has a central role in regulating inflammasome activation and subsequent IL-1β secretion [38, 90].

We also found over-representation of the oxygen-dependent proline HIF1A pathway and up-regulation of *HIF1A* (transcription factor HIF1α)[91]. Our findings were consistent with studies showing significant overexpression of *HIF1A* during EOS in preterm infants (<32 weeks gestation) [92] and during bacterial sepsis in very preterm infants [16]. The induction of HIF1α in response to sepsis-associated cytopathic hypoxia has also been shown in previous human adult and mouse sepsis studies [33, 93–96]. Importantly, HIF1α plays a crucial role in the metabolic switch from oxidative phosphorylation to glycolysis to meet increased energy demands for inflammatory responses during sepsis [33, 91, 97]. Further, HIF1α induces genes for proteins associated with glycolytic processes including lactate dehydrogenase (*LDHA*) and MCT4 (*SLC16A3*) [33, 91], where both genes were up-regulated in infants with confirmed LOS. Therefore, consistent with previous sepsis studies [16, 33, 91, 97, 98], our findings suggest that LOS is also associated with a shift towards glycolysis to meet the increased energy requirements for host immune responses in very preterm infants.

Finally, we found that host immune responses of infants with possible LOS could not be clearly distinguished from those with either confirmed LOS or no LOS–there were no differentially expressed genes between infants with possible LOS or no LOS, and only 21 differentially expressed genes when compared to infants with confirmed LOS. The small sample size of infants with possible LOS could potentially contribute to the lack of differentially expressed genes identified in comparison to infants with confirmed and no LOS. However, at the protein-level, infants with possible LOS did have significantly higher plasma levels of IL-6 and IL-10 compared to infants with no LOS, and similar levels to infants with confirmed LOS. Infants with possible LOS were identified by their elevated CRP levels in the absence of positive blood cultures. The production of plasma CRP by hepatocytes is controlled in part by IL-6 [44] and consistent with a study of preterm infants with suspected sepsis, our findings are consistent with those showing a positive relationship between CRP and IL-6 plasma levels [45]. We identified several genes associated with IL-6 (e.g. *NFKBIA*, *MYD88*, *IRAK3* and *IRKA4*) and IL-10 (e.g. *IL10RA* and *IL10*) production, although these were not significantly different in infants with possible LOS compared to those with no LOS which could be due to the differential expression criteria used and the small sample size of infants with possible LOS. The septic state of infants with possible LOS among a population of infants with suspected LOS remains challenging due to ambiguous host responses and examination of additional systems-level responses, such as at the metabolic levels, could allow us to better comprehend the underlying functional biology associated with possible LOS [99–101].

## Limitations

The limitations of our study included small sample sizes with no formal power calculations and no inclusion of the severity of illness score to describe preterm infants with LOS within this analysis [102], however, we have begun collecting this information and will report it in future studies. Our study demonstrates the ability to use RNA from small blood volumes (less than 0.5 mL) for transcriptional profiling at the systems-level using an unbiased RNA-Seq approach. The capacity to stabilise nucleic acids at the time of sample collection from low volumes of blood is also clinically relevant due to the difficulties associated with obtaining large blood volumes from neonates, especially those born preterm [16].

## Conclusion

We found that confirmed LOS is characterised by host immune transcriptional responses that reflect unbalanced immunometabolic homeostasis. The findings of this study represent a starting point for systems-level investigations of neonatal sepsis in very preterm infants and warrant validation in a statistically powered study. Investigation of LOS pathophysiology at the systems-level could lead to identification of novel gene signatures and gene products that could improve diagnosis of neonatal sepsis.

## Supporting information

**S1 Table. Metadata of study participants.**
(CSV)

**S2 Table. Differentially expressed gene list between confirmed LOS and no LOS groups.**
(CSV)

**S3 Table. Differentially expressed gene list between confirmed LOS and possible LOS groups.**
(CSV)

**S4 Table. Differential cell count analysis between clinical groups.**
(DOCX)

**S5 Table. Genes associated with over-represented pathways identified using the Sigora gene-pair signature method.**
(DOCX)

## Acknowledgments

The authors thank the research nurses and doctors at King Edward Memorial Hospital, Perth, Australia, for recruiting and collecting patient samples. The authors express heartfelt thanks to the infants and their families for their participation in this study.

## Author Contributions

**Conceptualization:** Sherrianne Ng, Tobias Strunk, Andrew Currie.

**Data curation:** Sherrianne Ng, Amy H. Lee.

**Formal analysis:** Sherrianne Ng, Amy H. Lee, Erin E. Gill, Reza Falsafi.

**Investigation:** Sherrianne Ng, Tabitha Woodman.

**Methodology:** Sherrianne Ng, Amy H. Lee, Erin E. Gill, Reza Falsafi, Julie Hibbert.

**Project administration:** Tobias Strunk, Julie Hibbert, Robert E. W. Hancock, Andrew Currie.

**Resources:** Robert E. W. Hancock, Andrew Currie.

**Supervision:** Tobias Strunk, Amy H. Lee, Robert E. W. Hancock, Andrew Currie.

**Writing – original draft:** Sherrianne Ng.

**Writing – review & editing:** Sherrianne Ng, Tobias Strunk, Amy H. Lee, Erin E. Gill, Reza Falsafi, Julie Hibbert, Robert E. W. Hancock, Andrew Currie.

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
