## [Decision Letter · Decision Letter 0]

13 Feb 2020

PONE-D-19-34983

Whole blood transcriptional responses of very preterm infants during late-onset sepsis

PLOS ONE

Dear Dr. Currie,

Thank you for submitting your manuscript to PLOS ONE. After careful consideration, we feel that it has merit but does not fully meet PLOS ONE’s publication criteria as it currently stands. Therefore, we invite you to submit a revised version of the manuscript that addresses the points raised during the review process.

According to referees' suggestions (see detailed comments below), the manuscript should be improved including de data and lists of differentially expressed genes. In addition, some methodological aspects should be clarified, and the interpretation and discussion of the obtained results accordingly explained. 

We would appreciate receiving your revised manuscript by Mar 28 2020 11:59PM. To enhance the reproducibility of your results, we recommend that if applicable you deposit your laboratory protocols in protocols.io, where a protocol can be assigned its own identifier (DOI) such that it can be cited independently in the future. For instructions see: http://journals.plos.org/plosone/s/submission-guidelines#loc-laboratory-protocols

We look forward to receiving your revised manuscript.

Kind regards,

Francisco J. Esteban, Ph.D., M.Sc.

Academic Editor

PLOS ONE

Journal Requirements:

3. We note you have a related manuscript currently under consideration at PLOS ONE. To comply with our policy (http://journals.plos.org/plosone/s/ethical-publishing-practice#loc-submission-and-publication-of-related-studies), please upload a copy of the related manuscript (PONE-D-19-33625, “Plasma cytokine profiles in very preterm infants with late-onset sepsis”) as 'other' files.

Reviewers' comments:

Reviewer's Responses to Questions

**Comments to the Author**

1. Is the manuscript technically sound, and do the data support the conclusions?

Reviewer #1: Partly

Reviewer #2: No

2. Has the statistical analysis been performed appropriately and rigorously? 

Reviewer #1: Yes

Reviewer #2: Yes

3. Have the authors made all data underlying the findings in their manuscript fully available?

Reviewer #1: Yes

Reviewer #2: No

4. Is the manuscript presented in an intelligible fashion and written in standard English?

Reviewer #1: Yes

Reviewer #2: Yes

5. Review Comments to the Author

Reviewer #1: In this manuscript, Dr. Ng and Colleagues used RNAseq to interrogate the whole blood response in preterm neonates with and without late-onset sepsis (LOS). The group has published several excellent studies on neonates, immunology, inflammation, and sepsis. There have been several studies of this type (whole blood RNA, neonate, sepsis) but to my knowledge this is the first to describe using RNAseq. They concluded host immune transcriptional responses that reflect unbalanced immunometabolic homeostasis.

As the authors are well aware, synthesis of neonatal sepsis studies is impaired by the lack of a consensus definition for sepsis in this population (PMID: 24751791).

There is a severity of illness score specific to preterm neonates with LOS that might be beneficial to include at the time of sepsis evaluation for the patients described herein (PMID: 31394566) so that the results can be interpreted in the context of future studies.

The median birth GA of the patients was very similar across groups but the birth weights were substantially different. Would the authors comment on this?

Would it be correct to assume no neonate died with the sepsis episode? Would it also be true that those without LOS received <=48 hrs abx? For infants that were in the possible LOS group - would it be true that they all received treatment for >5 days similar to those with a positive blood culture?

In the methods under RNAseq there is a statement that the samples were pooled. Later in the results there is a description of individual results. This is confusing and would benefit from clarification. If pooled, pooling may come with limitations that perhaps should be in the discussion (PMID: 26208977).

Some pathogens isolated from blood are unusual based on published large epidemiologic studies. Do the authors suspect that these unusual pathogens may have affected the applicability of the results?

The types of cells in whole blood can modify the transcriptome. Were CBC assessments done for clinical reasons in proximity to the RNA assessments that might inform the analysis? Consider including weeks of post-conceptual age alongside days on pg9 (line 158).

The data presented on pg 10 lines 163-168 is a bit confusing (confirmed v no sepsis = 1317 genes; confirmed v possible = 21 genes; possible v no sepsis = no differences). Would the authors clarify how the possible sepsis group can be so similar to both the confirmed bacteremic group and the no sepsis group?

IL-7 may indeed be beneficial to reduce adaptive impairment in sepsis. However a study of neonatal mice that lacked an adaptive immune system showed no difference in sepsis survival (PMID: 18591384) suggesting the role of adaptive immunity may be different in neonates compared to adults.

The discussion is very thorough but perhaps a bit long.

Reviewer #2: The article by Ng et al. is interesting and important as it covers a much understudied area of developing next-generation biomarkers for sepsis in most at-risk neonates and deciphering potential causal molecular pathways using relatively unbiased approached.

However, the article has two major weaknesses that I think are absolutely essential to address in order to bring it to sufficient publication quality for PLoSOne:

1) Much of the discussion is based on differentially expressed genes, based on data that are not presented in the manuscript! Or at least it could possible that the list of differentially expressed genes (including estimates and adjusted significance measures) have been included with the submission but in any case I could not see these data. It is not sufficient to say that there were so many differentially expressed genes between groups, or that this or that gene was differentially expressed. The data has to be presented quantitatively. Please make sure it is included in the next submission.

2) The manuscript goes on at length on negative findings, ignoring that this study is of extremely small sample size. Therefore, though I agree that the findings are original and worthy of publication the entire discussion has to be revised with considering this major limitation (I acknowledge that they list this as a limitation at the end of the discussion but they don’t sufficiently take this into consideration in their interpretation of data). Three examples:

a. Page 15, lines 291-293: the fact that infants with possible LOS had no differentially expressed genes compared to infants without LOS at the transcriptional-level may very well due to underpower and thus cannot be a major finding. To have this statement be a major finding would imply at least proper demonstration of power.

b. Same for the extended discussion around the lack of difference in IL-1b which may follow a trend but is clearly underpowered.

c. Page 20, lines 402: The possibility that many genes may not show up as differentially expressed needs to be considered here.

In addition, I have a number of other suggestions for improvement (ordered by appearance in text):

Introduction :

“Neonatal sepsis results in more than 450,000 infant deaths globally each year.” Since these are only gross estimates, based on extrapolated data, I think it should be reworded. Consider including a more recent reference than the Lawn J, such as this one latest one: KE Rudd, Lancet 2020 who report a potential for 801,615 neonatal deaths per year due to sepsis. Probably the “real” number is between half million to one million neonatal deaths due to sepsis each year.

“In addition, preterm infants have also shown impairments in cellular metabolism that limit 40 innate immune responses during infection [11].” Please clarify. I think you mean that immune cells from preterm infants, not the preterm infants themselves…”.

“However, no study to date has specifically investigated host transcriptional responses during LOS in very preterm infants, in whom the incidence of sepsis is highest.” “Our findings are the first to characterise the underlying functional biology of blood leukocyte transcriptional responses associated specifically with LOS immunopathology in very preterm infants. “This is not entirely true as the author point out the study by Cernada was conducted in VLBW infants. Please adjust these 2 statements and better emphasize how this study builds on Cernada’s findings in the introduction. This is crucial to increase the value of the authors’ work relatively to what has already been published.

Methods:

Please indicate the recommended blood culture volume and methods for infants in the study NICU and if available, how much blood was actually sent to the lab for infants in the study. Please also indicate what CRP assay was used in the study NICU at the time of the study period.

The rationale for the fold-change cut-off is unclear to me. Given the small dataset I would have only used an adjusted p value cut-off (the fold-change probably adds little and is somewhat arbitrary).

Results:

Please provide the clinical data in non-aggregated form for all subjects in 3 group (not just septic babies) in a supplementary table, This will be crucial to allow future linkage by study ID to the transcriptome data submitted to GEO and help build on the author’s finding in future research.

The authors need to provide a list of differentially expressed genes with adjusted p values and effect size, between each group comparisons, in order to substantiate the core of their findings.

Can the author provide data on the blood count levels for the infants at the time of blood sampling for RNA?

Discussion:

There are several comments made about the lack of secretion cytokines in these infants, although for cytokines like interferons (alpha or gamma) there is very little published report that these are sufficiently detectable levels in human blood and this may very well be due to the fact that these cytokines act locally in their micro-environment rather than systemically. I think the choice of measured cytokines need to be better justified and the comment on the lack of detection tone down, OR there needs to be strong literature support of the feasibility of detecting these cytokines, for example, in adults with sepsis (as a sort of “historical” positive control).

Page 17, lines 338-339: In addition to my comment above, about the lack of detectable IL-1b and justification for measuring this cytokine (for interpretation of negative finding). I wonder if the authors have considered that the alternative explanation that the lack of IL-1b (assuming it is a detectable cytokine) could be related to a weak cytokine response predisposing preterm infants to sepsis rather than an active inhibition via IL1R2/IL1RN.

I find the discussion around metabolism (particularly cholesterol but also basic energy metabolism) quite relevant and interesting. Therefore I suggest brining that up in the discussion and emphasizing this point. As the authors point out there is an exciting literature in adults with sepsis related to cholesterol metabolism for prognostication, the use of lipid proteins inducers and role in neutralization of LPS.

SOCS1/SOCS3: without seeing the actual data (quantitative measures) it is very difficult to appreciate this section which seems a lot more speculative (in view also with the strong possibility that IFN may not be measurable meaningfully in blood).

It would be informative to provide blood counts for these infants. In addition, the authors could estimate these counts using hematopoietic cell-specific markers. The possibility that differences in cell population is not sufficiently considered in the interpretation of the data. Estimating blood counts using in silico methods would strongly help such interpretation of the data where relevant.

Page 20, lines 402-417: There is a major obvious consideration that is missing here in this part of the discussion, which is the fact that among babies with possible sepsis there is likely a number that actually don’t have bacterial sepsis, but have non-bacterial sepsis or other inflammatory syndromes. Have the authors tried to apply their top differentially expressed genes from the confirmed sepsis group to try to separate this group further. This may be futile given the small sample size. At least discuss this possibility. At the moment, the paragraph is quite speculative in my view and the lack of power needs to be better considered. Were there any genes that were very close to the adjusted p value threshold? Impossible to say without seeing the data and an estimation of power needs to support the currently exposed speculations.

6. PLOS authors have the option to publish the peer review history of their article (what does this mean?). If published, this will include your full peer review and any attached files.

Reviewer #1: No

Reviewer #2: No

---

## [Author Response · Author response to Decision Letter 0]

23 Apr 2020

A detailed response to all reviewers' comments has been attached as a separate document.

---

## [Decision Letter · Decision Letter 1]

14 May 2020

Whole blood transcriptional responses of very preterm infants during late-onset sepsis

PONE-D-19-34983R1

Dear Dr. Currie,

We are pleased to inform you that your manuscript has been judged scientifically suitable for publication and will be formally accepted for publication once it complies with all outstanding technical requirements.

With kind regards,

Francisco J. Esteban, Ph.D., M.Sc.

Academic Editor

PLOS ONE

Additional Editor Comments (optional):

According to reviewer#2: Excellent work addressing the comments. Please recheck the supplemental data provided: S1 Table appears duplicated with one of the S1 Table being identical to S5 Table.

Reviewers' comments:

Reviewer's Responses to Questions

**Comments to the Author**

1. If the authors have adequately addressed your comments raised in a previous round of review and you feel that this manuscript is now acceptable for publication, you may indicate that here to bypass the “Comments to the Author” section, enter your conflict of interest statement in the “Confidential to Editor” section, and submit your "Accept" recommendation.

Reviewer #1: All comments have been addressed

Reviewer #2: All comments have been addressed

2. Is the manuscript technically sound, and do the data support the conclusions?

Reviewer #1: Yes

Reviewer #2: Yes

3. Has the statistical analysis been performed appropriately and rigorously? 

Reviewer #1: Yes

Reviewer #2: Yes

4. Have the authors made all data underlying the findings in their manuscript fully available?

Reviewer #1: Yes

Reviewer #2: Yes

5. Is the manuscript presented in an intelligible fashion and written in standard English?

Reviewer #1: Yes

Reviewer #2: Yes

6. Review Comments to the Author

Reviewer #1: Thank you for your responses to my queries. ________________________________________________________

Reviewer #2: Excellent work addressing the comments. Please recheck the supplemental data provided: S1 Table appears duplicated with one of the S1 Table being identical to S5 Table.

7. PLOS authors have the option to publish the peer review history of their article (what does this mean?). If published, this will include your full peer review and any attached files.

Reviewer #1: No

Reviewer #2: Yes: Dr. Pascal Lavoie, University of British Columbia & BC Children's Hospital Research Institute, Vancouver Canada

---

## [Editor Report · Acceptance letter]

20 May 2020

PONE-D-19-34983R1 

Whole blood transcriptional responses of very preterm infants during late-onset sepsis 

Dear Dr. Currie:

I am pleased to inform you that your manuscript has been deemed suitable for publication in PLOS ONE. Congratulations! Your manuscript is now with our production department. 

With kind regards,

on behalf of

Dr. Francisco J. Esteban 

Academic Editor

PLOS ONE